# The Poisson Gamma Belief Network

**Mingyuan Zhou**
McCombs School of Business
The University of Texas at Austin
Austin, TX 78712, USA

**Yulai Cong**
National Laboratory of RSP
Xidian University
Xi'an, Shaanxi, China

**Bo Chen**
National Laboratory of RSP
Xidian University
Xi'an, Shaanxi, China

## Abstract

To infer a multilayer representation of high-dimensional count vectors, we propose the Poisson gamma belief network (PGBN) that factorizes each of its layers into the product of a connection weight matrix and the nonnegative real hidden units of the next layer. The PGBN's hidden layers are jointly trained with an upward-downward Gibbs sampler, each iteration of which upward samples Dirichlet distributed connection weight vectors starting from the first layer (bottom data layer), and then downward samples gamma distributed hidden units starting from the top hidden layer. The gamma-negative binomial process combined with a layer-wise training strategy allows the PGBN to infer the width of each layer given a fixed budget on the width of the first layer. The PGBN with a single hidden layer reduces to Poisson factor analysis. Example results on text analysis illustrate interesting relationships between the width of the first layer and the inferred network structure, and demonstrate that the PGBN, whose hidden units are imposed with correlated gamma priors, can add more layers to increase its performance gains over Poisson factor analysis, given the same limit on the width of the first layer.

## 1 Introduction

There has been significant recent interest in deep learning. Despite its tremendous success in supervised learning, inferring a multilayer data representation in an unsupervised manner remains a challenging problem [1, 2, 3]. The sigmoid belief network (SBN), which connects the binary units of adjacent layers via the sigmoid functions, infers a deep representation of multivariate binary vectors [4, 5]. The deep belief network (DBN) [6] is a SBN whose top hidden layer is replaced by the restricted Boltzmann machine (RBM) [7] that is undirected. The deep Boltzmann machine (DBM) is an undirected deep network that connects the binary units of adjacent layers using the RBMs [8]. All these deep networks are designed to model binary observations. Although one may modify the bottom layer to model Gaussian and multinomial observations, the hidden units of these networks are still typically restricted to be binary [8, 9, 10]. One may further consider the exponential family harmoniums [11, 12] to construct more general networks with non-binary hidden units, but often at the expense of noticeably increased complexity in training and data fitting.

Moving beyond conventional deep networks using binary hidden units, we construct a deep directed network with gamma distributed nonnegative real hidden units to unsupervisedly infer a multilayer representation of multivariate count vectors, with a simple but powerful mechanism to capture the correlations among the visible/hidden features across all layers and handle highly overdispersed counts. The proposed model is called the Poisson gamma belief network (PGBN), which factorizes the observed count vectors under the Poisson likelihood into the product of a factor loading matrix and the gamma distributed hidden units (factor scores) of layer one; and further factorizes the shape parameters of the gamma hidden units of each layer into the product of a connection weight matrix and the gamma hidden units of the next layer. Distinct from previous deep networks that often utilize binary units for tractable inference and require tuning both the width (number of hidden units) of each layer and the network depth (number of layers), the PGBN employs nonnegative real hidden

units and automatically infers the widths of subsequent layers given a fixed budget on the width of its first layer. Note that the budget could be infinite and hence the whole network can grow without bound as more data are being observed. When the budget is finite and hence the ultimate capacity of the network is limited, we find that the PGBN equipped with a narrower first layer could increase its depth to match or even outperform a shallower network with a substantially wider first layer.

The gamma distribution density function has the highly desired strong non-linearity for deep learning, but the existence of neither a conjugate prior nor a closed-form maximum likelihood estimate for its shape parameter makes a deep network with gamma hidden units appear unattractive. Despite seemingly difficult, we discover that, by generalizing the data augmentation and marginalization techniques for discrete data [13], one may propagate latent counts one layer at a time from the bottom data layer to the top hidden layer, with which one may derive an efficient upward-downward Gibbs sampler that, one layer at a time in each iteration, upward samples Dirichlet distributed connection weight vectors and then downward samples gamma distributed hidden units.

In addition to constructing a new deep network that well fits multivariate count data and developing an efficient upward-downward Gibbs sampler, other contributions of the paper include: 1) combining the gamma-negative binomial process [13, 14] with a layer-wise training strategy to automatically infer the network structure; 2) revealing the relationship between the upper bound imposed on the width of the first layer and the inferred widths of subsequent layers; 3) revealing the relationship between the network depth and the model's ability to model overdispersed counts; 4) and generating a multivariate high-dimensional random count vector, whose distribution is governed by the PGBN, by propagating the gamma hidden units of the top hidden layer back to the bottom data layer.

## 1.1 Useful count distributions and their relationships

Let the Chinese restaurant table (CRT) distribution $l \sim \mathrm{CRT}(n, r)$ represent the distribution of a random count generated as $l = \sum_{i=1}^{n} b_i$, $b_i \sim \mathrm{Bernoulli}\left[r/(r + i - 1)\right]$. Its probability mass function (PMF) can be expressed as $P(l \mid n, r) = \frac{\Gamma(r) r^l}{\Gamma(n+r)} |s(n, l)|$, where $l \in \mathbb{Z}$, $\mathbb{Z} := \{0, 1, \ldots, n\}$, and $|s(n, l)|$ are unsigned Stirling numbers of the first kind. Let $u \sim \mathrm{Log}(p)$ denote the logarithmic distribution with PMF $P(u \mid p) = \frac{1}{-\ln(1-p)} \frac{p^u}{u}$, where $u \in \{1, 2, \ldots\}$. Let $n \sim \mathrm{NB}(r, p)$ denote the negative binomial (NB) distribution with PMF $P(n \mid r, p) = \frac{\Gamma(n+r)}{n! \Gamma(r)} p^n (1 - p)^r$, where $n \in \mathbb{Z}$. The NB distribution $n \sim \mathrm{NB}(r, p)$ can be generated as a gamma mixed Poisson distribution as $n \sim \mathrm{Pois}(\lambda)$, $\lambda \sim \mathrm{Gam}\left[r, p/(1-p)\right]$, where $p/(1-p)$ is the gamma scale parameter. As shown in [13], the joint distribution of $n$ and $l$ given $r$ and $p$ in $l \sim \mathrm{CRT}(n, r)$, $n \sim \mathrm{NB}(r, p)$, where $l \in \{0, \ldots, n\}$ and $n \in \mathbb{Z}$, is the same as that in $n = \sum_{t=1}^{l} u_t$, $u_t \sim \mathrm{Log}(p)$, $l \sim \mathrm{Pois}[-r \ln(1 - p)]$, which is called the Poisson-logarithmic bivariate distribution, with PMF $P(n, l \mid r, p) = \frac{|s(n,l)| r^l}{n!} p^n (1 - p)^r$.

## 2 The Poisson Gamma Belief Network

Assuming the observations are multivariate count vectors $\boldsymbol{x}_j^{(1)} \in \mathbb{Z}^{K_0}$, the generative model of the Poisson gamma belief network (PGBN) with $T$ hidden layers, from top to bottom, is expressed as

$$\boldsymbol{\theta}_j^{(T)} \sim \mathrm{Gam}\left(\boldsymbol{r}, 1/c_j^{(T+1)}\right),$$
$$\cdots$$
$$\boldsymbol{\theta}_j^{(t)} \sim \mathrm{Gam}\left(\boldsymbol{\Phi}^{(t+1)} \boldsymbol{\theta}_j^{(t+1)}, 1/c_j^{(t+1)}\right),$$
$$\cdots$$
$$\boldsymbol{x}_j^{(1)} \sim \mathrm{Pois}\left(\boldsymbol{\Phi}^{(1)} \boldsymbol{\theta}_j^{(1)}\right), \quad \boldsymbol{\theta}_j^{(1)} \sim \mathrm{Gam}\left(\boldsymbol{\Phi}^{(2)} \boldsymbol{\theta}_j^{(2)}, p_j^{(2)}/\left(1 - p_j^{(2)}\right)\right). \tag{1}$$

The PGBN factorizes the count observation $\boldsymbol{x}_j^{(1)}$ into the product of the factor loading $\boldsymbol{\Phi}^{(1)} \in \mathbb{R}_+^{K_0 \times K_1}$ and hidden units $\boldsymbol{\theta}_j^{(1)} \in \mathbb{R}_+^{K_1}$ of layer one under the Poisson likelihood, where $\mathbb{R}_+ = \{x : x \geq 0\}$, and for $t = 1, 2, \ldots, T-1$, factorizes the shape parameters of the gamma distributed hidden units $\boldsymbol{\theta}_j^{(t)} \in \mathbb{R}_+^{K_t}$ of layer $t$ into the product of the connection weight matrix $\boldsymbol{\Phi}^{(t+1)} \in \mathbb{R}_+^{K_t \times K_{t+1}}$ and the hidden units $\boldsymbol{\theta}_j^{(t+1)} \in \mathbb{R}_+^{K_{t+1}}$ of layer $t + 1$; the top layer's hidden units $\boldsymbol{\theta}_j^{(T)}$ share the same

vector $\boldsymbol{r} = (r_1, \ldots, r_{K_T})'$ as their gamma shape parameters; and the $p_j^{(2)}$ are probability parameters and $\{1/c^{(t)}\}_{3,T+1}$ are gamma scale parameters, with $c_j^{(2)} := \big(1 - p_j^{(2)}\big)/p_j^{(2)}$.

For scale identifiabilty and ease of inference, each column of $\boldsymbol{\Phi}^{(t)} \in \mathbb{R}_+^{K_{t-1} \times K_t}$ is restricted to have a unit $L_1$ norm. To complete the hierarchical model, for $t \in \{1, \ldots, T-1\}$, we let

$$\boldsymbol{\phi}_k^{(t)} \sim \text{Dir}\big(\eta^{(t)}, \ldots, \eta^{(t)}\big), \quad r_k \sim \text{Gam}\big(\gamma_0/K_T, 1/c_0\big) \tag{2}$$

and impose $c_0 \sim \text{Gam}(e_0, 1/f_0)$ and $\gamma_0 \sim \text{Gam}(a_0, 1/b_0)$; and for $t \in \{3, \ldots, T+1\}$, we let

$$p_j^{(2)} \sim \text{Beta}(a_0, b_0), \quad c_j^{(t)} \sim \text{Gam}(e_0, 1/f_0). \tag{3}$$

We expect the correlations between the rows (features) of $(\boldsymbol{x}_1^{(1)}, \ldots, \boldsymbol{x}_J^{(1)})$ to be captured by the columns of $\boldsymbol{\Phi}^{(1)}$, and the correlations between the rows (latent features) of $(\boldsymbol{\theta}_1^{(t)}, \ldots, \boldsymbol{\theta}_J^{(t)})$ to be captured by the columns of $\boldsymbol{\Phi}^{(t+1)}$. Even if all $\boldsymbol{\Phi}^{(t)}$ for $t \geq 2$ are identity matrices, indicating no correlations between latent features, our analysis will show that a deep structure with $T \geq 2$ could still benefit data fitting by better modeling the variability of the latent features $\boldsymbol{\theta}_j^{(1)}$.

**Sigmoid and deep belief networks.** Under the hierarchical model in (1), given the connection weight matrices, the joint distribution of the count observations and gamma hidden units of the PGBN can be expressed, similar to those of the sigmoid and deep belief networks [3], as

$$P\left(\boldsymbol{x}_j^{(1)}, \{\boldsymbol{\theta}_j^{(t)}\}_t \,\middle|\, \{\boldsymbol{\Phi}^{(t)}\}_t\right) = P\left(\boldsymbol{x}_j^{(1)} \,\middle|\, \boldsymbol{\Phi}^{(1)}, \boldsymbol{\theta}_j^{(1)}\right)\left[\prod_{t=1}^{T-1} P\left(\boldsymbol{\theta}_j^{(t)} \,\middle|\, \boldsymbol{\Phi}^{(t+1)}, \boldsymbol{\theta}_j^{(t+1)}\right)\right] P\left(\boldsymbol{\theta}_j^{(T)}\right).$$

With $\boldsymbol{\phi}_{v:}$ representing the $v$th row $\boldsymbol{\Phi}$, for the gamma hidden units $\theta_{vj}^{(t)}$ we have

$$P\left(\theta_{vj}^{(t)} \,\middle|\, \boldsymbol{\phi}_{v:}^{(t+1)}, \boldsymbol{\theta}_j^{(t+1)}, c_{j+1}^{(t+1)}\right) = \frac{\big(c_{j+1}^{(t+1)}\big)^{\boldsymbol{\phi}_{v:}^{(t+1)}\boldsymbol{\theta}_j^{(t+1)}}}{\Gamma\big(\boldsymbol{\phi}_{v:}^{(t+1)}\boldsymbol{\theta}_j^{(t+1)}\big)} \left(\theta_{vj}^{(t)}\right)^{\boldsymbol{\phi}_{v:}^{(t+1)}\boldsymbol{\theta}_j^{(t+1)} - 1} e^{-c_{j+1}^{(t+1)}\theta_{vj}^{(t)}}, \quad (4)$$

which are highly nonlinear functions that are strongly desired in deep learning. By contrast, with the sigmoid function $\sigma(x) = 1/(1 + e^{-x})$ and bias terms $b_v^{(t+1)}$, a sigmoid/deep belief network would connect the binary hidden units $\theta_{vj}^{(t)} \in \{0, 1\}$ of layer $t$ (for deep belief networks, $t < T - 1$) to the product of the connection weights and binary hidden units of the next layer with

$$P\left(\theta_{vj}^{(t)} = 1 \,\middle|\, \boldsymbol{\phi}_{v:}^{(t+1)}, \boldsymbol{\theta}_j^{(t+1)}, b_v^{(t+1)}\right) = \sigma\left(b_v^{(t+1)} + \boldsymbol{\phi}_{v:}^{(t+1)}\boldsymbol{\theta}_j^{(t+1)}\right). \tag{5}$$

Comparing (4) with (5) clearly shows the differences between the gamma nonnegative hidden units and the sigmoid link based binary hidden units. Note that the rectified linear units have emerged as powerful alternatives of sigmoid units to introduce nonlinearity [15]. It would be interesting to use the gamma units to introduce nonlinearity in the positive region of the rectified linear units.

**Deep Poisson factor analysis.** With $T = 1$, the PGBN specified by (1)-(3) reduces to Poisson factor analysis (PFA) using the (truncated) gamma-negative binomial process [13], which is also related to latent Dirichlet allocation [16] if the Dirichlet priors are imposed on both $\boldsymbol{\phi}_k^{(1)}$ and $\boldsymbol{\theta}_j^{(1)}$. With $T \geq 2$, the PGBN is related to the gamma Markov chain hinted by Corollary 2 of [13] and realized in [17], the deep exponential family of [18], and the deep PFA of [19]. Different from the PGBN, in [18], it is the gamma scale but not shape parameters that are chained and factorized; in [19], it is the correlations between binary topic usage indicators but not the full connection weights that are captured; and neither [18] nor [19] provide a principled way to learn the network structure. Below we break the PGBN of $T$ layers into $T$ related submodels that are solved with the same subroutine.

## 2.1 The propagation of latent counts and model properties

**Lemma 1** (Augment-and-conquer the PGBN). *With $p_j^{(1)} := 1 - e^{-1}$ and*

$$p_j^{(t+1)} := -\ln(1 - p_j^{(t)}) \Big/ \left[c_j^{(t+1)} - \ln(1 - p_j^{(t)})\right] \tag{6}$$

*for $t = 1, \ldots, T$, one may connect the observed (if $t = 1$) or some latent (if $t \geq 2$) counts $\boldsymbol{x}_j^{(t)} \in \mathbb{Z}^{K_{t-1}}$ to the product $\boldsymbol{\Phi}^{(t)}\boldsymbol{\theta}_j^{(t)}$ at layer $t$ under the Poisson likelihood as*

$$\boldsymbol{x}_j^{(t)} \sim \text{Pois}\left[-\boldsymbol{\Phi}^{(t)}\boldsymbol{\theta}_j^{(t)} \ln\left(1 - p_j^{(t)}\right)\right]. \tag{7}$$

*Proof.* By definition (7) is true for layer $t = 1$. Suppose that (7) is true for layer $t \geq 2$, then we can augment each count $x_{vj}^{(t)}$ into the summation of $K_t$ latent counts that are smaller or equal as

$$x_{vj}^{(t)} = \sum_{k=1}^{K_t} x_{vjk}^{(t)}, \quad x_{vjk}^{(t)} \sim \text{Pois}\left[-\phi_{vk}^{(t)}\theta_{kj}^{(t)}\ln\left(1 - p_j^{(t)}\right)\right], \tag{8}$$

where $v \in \{1, \ldots, K_{t-1}\}$. With $m_{kj}^{(t)(t+1)} := x_{\cdot jk}^{(t)} := \sum_{v=1}^{K_{t-1}} x_{vjk}^{(t)}$ representing the number of times that factor $k \in \{1, \ldots, K_t\}$ of layer $t$ appears in observation $j$ and $\boldsymbol{m}_j^{(t)(t+1)} := \left(x_{\cdot j1}^{(t)}, \ldots, x_{\cdot jK_t}^{(t)}\right)'$, since $\sum_{v=1}^{K_{t-1}} \phi_{vk}^{(t)} = 1$, we can marginalize out $\boldsymbol{\Phi}^{(t)}$ as in [20], leading to

$$\boldsymbol{m}_j^{(t)(t+1)} \sim \text{Pois}\left[-\boldsymbol{\theta}_j^{(t)}\ln\left(1 - p_j^{(t)}\right)\right].$$

Further marginalizing out the gamma distributed $\boldsymbol{\theta}_j^{(t)}$ from the above Poisson likelihood leads to

$$\boldsymbol{m}_j^{(t)(t+1)} \sim \text{NB}\left(\boldsymbol{\Phi}^{(t+1)}\boldsymbol{\theta}_j^{(t+1)}, p_j^{(t+1)}\right). \tag{9}$$

The $k$th element of $\boldsymbol{m}_j^{(t)(t+1)}$ can be augmented under its compound Poisson representation as

$$m_{kj}^{(t)(t+1)} = \sum_{\ell=1}^{x_{kj}^{(t+1)}} u_\ell, \quad u_\ell \sim \text{Log}(p_j^{(t+1)}), \quad x_{kj}^{(t+1)} \sim \text{Pois}\left[-\phi_{k:}^{(t+1)}\boldsymbol{\theta}_j^{(t+1)}\ln\left(1 - p_j^{(t+1)}\right)\right].$$

Thus if (7) is true for layer $t$, then it is also true for layer $t + 1$. $\qquad\square$

**Corollary 2** (Propagate the latent counts upward). *Using Lemma 4.1 of [20] on (8) and Theorem 1 of [13] on (9), we can propagate the latent counts $x_{vj}^{(t)}$ of layer $t$ upward to layer $t + 1$ as*

$$\left\{\left(x_{vj1}^{(t)}, \ldots, x_{vjK_t}^{(t)}\right) \mid x_{vj}^{(t)}, \boldsymbol{\phi}_{v:}^{(t)}, \boldsymbol{\theta}_j^{(t)}\right\} \sim \text{Mult}\left(x_{vj}^{(t)}, \frac{\phi_{v1}^{(t)}\theta_{1j}^{(t)}}{\sum_{k=1}^{K_t}\phi_{vk}^{(t)}\theta_{kj}^{(t)}}, \ldots, \frac{\phi_{vK_t}^{(t)}\theta_{K_t j}^{(t)}}{\sum_{k=1}^{K_t}\phi_{vk}^{(t)}\theta_{kj}^{(t)}}\right), \tag{10}$$

$$\left(x_{kj}^{(t+1)} \mid m_{kj}^{(t)(t+1)}, \boldsymbol{\phi}_{k:}^{(t+1)}, \boldsymbol{\theta}_j^{(t+1)}\right) \sim \text{CRT}\left(m_{kj}^{(t)(t+1)}, \boldsymbol{\phi}_{k:}^{(t+1)}\boldsymbol{\theta}_j^{(t+1)}\right). \tag{11}$$

As $x_{\cdot j}^{(t)} = m_{\cdot j}^{(t)(t+1)}$ and $x_{kj}^{(t+1)}$ is in the same order as $\ln\left(m_{kj}^{(t)(t+1)}\right)$, the total count of layer $t + 1$, expressed as $\sum_j x_{\cdot j}^{(t+1)}$, would often be much smaller than that of layer $t$, expressed as $\sum_j x_{\cdot j}^{(t)}$. Thus the PGBN may use $\sum_j x_{\cdot j}^{(T)}$ as a simple criterion to decide whether to add more layers.

## 2.2 Modeling overdispersed counts

In comparison to a single-layer shallow model with $T = 1$ that assumes the hidden units of layer one to be independent in the prior, the multilayer deep model with $T \geq 2$ captures the correlations between them. Note that for the extreme case that $\boldsymbol{\Phi}^{(t)} = \mathbf{I}_{K_t}$ for $t \geq 2$ are all identity matrices, which indicates that there are no correlations between the features of $\boldsymbol{\theta}_j^{(t-1)}$ left to be captured, the deep structure could still provide benefits as it helps model latent counts $\boldsymbol{m}_j^{(1)(2)}$ that may be highly overdispersed. For example, supposing $\boldsymbol{\Phi}^{(t)} = \mathbf{I}_{K_2}$ for all $t \geq 2$, then from (1) and (9) we have

$$m_{kj}^{(1)(2)} \sim \text{NB}(\theta_{kj}^{(2)}, p_j^{(2)}), \quad \ldots, \quad \theta_{kj}^{(t)} \sim \text{Gam}(\theta_{kj}^{(t+1)}, 1/c_j^{(t+1)}), \quad \ldots, \quad \theta_{kj}^{(T)} \sim \text{Gam}(r_k, 1/c_j^{(T+1)}).$$

For simplicity, let us further assume $c_j^{(t)} = 1$ for all $t \geq 3$. Using the laws of total expectation and total variance, we have $\mathbb{E}\left[\theta_{kj}^{(2)} \mid r_k\right] = r_k$ and $\text{Var}\left[\theta_{kj}^{(2)} \mid r_k\right] = (T - 1)r_k$, and hence

$$\mathbb{E}\left[m_{kj}^{(1)(2)} \mid r_k\right] = r_k p_j^{(2)}/(1 - p_j^{(2)}), \quad \text{Var}\left[m_{kj}^{(1)(2)} \mid r_k\right] = r_k p_j^{(2)}\left(1 - p_j^{(2)}\right)^{-2}\left[1 + (T - 1)p_j^{(2)}\right].$$

In comparison to PFA with $m_{kj}^{(1)(2)} \mid r_k \sim \text{NB}(r_k, p_j^{(2)})$, with a variance-to-mean ratio of $1/(1 - p_j^{(2)})$, the PGBN with $T$ hidden layers, which mixes the shape of $m_{kj}^{(1)(2)} \sim \text{NB}(\theta_{kj}^{(2)}, p_j^{(2)})$ with a chain of gamma random variables, increases the variance-to-mean ratio of the latent count $m_{kj}^{(1)(2)}$ given $r_k$ by a factor of $1 + (T - 1)p_j^{(2)}$, and hence could better model highly overdispersed counts.

## 2.3 Upward-downward Gibbs sampling

With Lemma 1 and Corollary 2 and the width of the first layer being bounded by $K_{1\max}$, we develop an upward-downward Gibbs sampler for the PGBN, each iteration of which proceeds as follows:

**Sample** $x_{vjk}^{(t)}$. We can sample $x_{vjk}^{(t)}$ for all layers using (10). But for the first hidden layer, we may treat each observed count $x_{vj}^{(1)}$ as a sequence of word tokens at the $v$th term (in a vocabulary of size $V := K_0$) in the $j$th document, and assign the $x_{\cdot j}^{(1)}$ words $\{v_{ji}\}_{i=1,x_{\cdot j}^{(1)}}$ one after another to the latent factors (topics), with both the topics $\boldsymbol{\Phi}^{(1)}$ and topic weights $\boldsymbol{\theta}_j^{(1)}$ marginalized out, as

$$P(z_{ji} = k \mid -) \propto \frac{\eta^{(1)} + x_{v_{ji}\cdot k}^{(1)-ji}}{V\eta^{(1)} + x_{\cdot\cdot k}^{(1)-ji}} \left( x_{\cdot jk}^{(1)-ji} + \boldsymbol{\phi}_{k:}^{(2)}\boldsymbol{\theta}_j^{(2)} \right), \quad k \in \{1, \ldots, K_{1\max}\}, \qquad (12)$$

where $z_{ji}$ is the topic index for $v_{ji}$ and $x_{vjk}^{(1)} := \sum_i \delta(v_{ji} = v, z_{ji} = k)$ counts the number of times that term $v$ appears in document $j$; we use the $\cdot$ symbol to represent summing over the corresponding index, *e.g.*, $x_{\cdot jk}^{(t)} := \sum_v x_{vjk}^{(t)}$, and use $x^{-ji}$ to denote the count $x$ calculated without considering word $i$ in document $j$. The collapsed Gibbs sampling update equation shown above is related to the one developed in [21] for latent Dirichlet allocation, and the one developed in [22] for PFA using the beta-negative binomial process. When $T = 1$, we would replace the terms $\boldsymbol{\phi}_{k:}^{(2)}\boldsymbol{\theta}_j^{(2)}$ with $r_k$ for PFA built on the gamma-negative binomial process [13] (or with $\alpha\pi_k$ for the hierarchical Dirichlet process latent Dirichlet allocation, see [23] and [22] for details), and add an additional term to account for the possibility of creating an additional topic [22]. For simplicity, in this paper, we truncate the nonparametric Bayesian model with $K_{1\max}$ factors and let $r_k \sim \text{Gam}(\gamma_0/K_{1\max}, 1/c_0)$ if $T = 1$.

**Sample** $\boldsymbol{\phi}_k^{(t)}$. Given these latent counts, we sample the factors/topics $\boldsymbol{\phi}_k^{(t)}$ as

$$(\boldsymbol{\phi}_k^{(t)} \mid -) \sim \text{Dir}\left( \eta^{(t)} + x_{1\cdot k}^{(t)}, \ldots, \eta^{(t)} + x_{K_{t-1}\cdot k}^{(t)} \right). \qquad (13)$$

**Sample** $x_{vj}^{(t+1)}$. We sample $\boldsymbol{x}_j^{(t+1)}$ using (11), replacing $\boldsymbol{\Phi}^{(T+1)}\boldsymbol{\theta}_j^{(T+1)}$ with $\boldsymbol{r} := (r_1, \ldots, r_{K_T})'$.

**Sample** $\boldsymbol{\theta}_j^{(t)}$. Using (7) and the gamma-Poisson conjugacy, we sample $\boldsymbol{\theta}_j$ as

$$(\boldsymbol{\theta}_j^{(t)} \mid -) \sim \text{Gamma}\left( \boldsymbol{\Phi}^{(t+1)}\boldsymbol{\theta}_j^{(t+1)} + \boldsymbol{m}_j^{(t)(t+1)}, \left[ c_j^{(t+1)} - \ln\left(1 - p_j^{(t)}\right) \right]^{-1} \right). \qquad (14)$$

**Sample** $\boldsymbol{r}$. Both $\gamma_0$ and $c_0$ are sampled using related equations in [13]. We sample $\boldsymbol{r}$ as

$$(r_v \mid -) \sim \text{Gam}\left( \gamma_0/K_T + x_{v\cdot}^{(T+1)}, \left[ c_0 - \sum_j \ln\left(1 - p_j^{(T+1)}\right) \right]^{-1} \right). \qquad (15)$$

**Sample** $c_j^{(t)}$. With $\theta_{\cdot j}^{(t)} := \sum_{k=1}^{K_t} \theta_{kj}^{(t)}$ for $t \leq T$ and $\theta_{\cdot j}^{(T+1)} := r_{\cdot}$, we sample $p_j^{(2)}$ and $\{c_j^{(t)}\}_{t\geq 3}$ as

$$(p_j^{(2)} \mid -) \sim \text{Beta}\left( a_0 + m_{\cdot j}^{(1)(2)}, b_0 + \theta_{\cdot j}^{(2)} \right), \quad (c_j^{(t)} \mid -) \sim \text{Gamma}\left( e_0 + \theta_{\cdot j}^{(t)}, \left[ f_0 + \theta_{\cdot j}^{(t-1)} \right]^{-1} \right), \quad (16)$$

and calculate $c_j^{(2)}$ and $\{p_j^{(t)}\}_{t\geq 3}$ with (6).

## 2.4 Learning the network structure with layer-wise training

As jointly training all layers together is often difficult, existing deep networks are typically trained using a greedy layer-wise unsupervised training algorithm, such as the one proposed in [6] to train the deep belief networks. The effectiveness of this training strategy is further analyzed in [24]. By contrast, the PGBN has a simple Gibbs sampler to jointly train all its hidden layers, as described in Section 2.3, and hence does not require greedy layer-wise training. Yet the same as commonly used deep learning algorithms, it still needs to specify the number of layers and the width of each layer.

In this paper, we adopt the idea of layer-wise training for the PGBN, not because of the lack of an effective joint-training algorithm, but for the purpose of learning the width of each hidden layer in a greedy layer-wise manner, given a fixed budget on the width of the first layer. The proposed layer-wise training strategy is summarized in Algorithm 1. With a PGBN of $T - 1$ layers that has already been trained, the key idea is to use a truncated gamma-negative binomial process [13] to model the latent count matrix for the newly added top layer as $m_{kj}^{(T)(T+1)} \sim \text{NB}(r_k, p_j^{(T+1)})$, $r_k \sim$

---

**Algorithm 1** The PGBN upward-downward Gibbs sampler that uses a layer-wise training strategy to train a set of networks, each of which adds an additional hidden layer on top of the previously inferred network, retrains all its layers jointly, and prunes inactive factors from the last layer. **Inputs:** observed counts $\{x_{vj}\}_{v,j}$, upper bound of the width of the first layer $K_{1\,\mathrm{max}}$, upper bound of the number of layers $T_{\mathrm{max}}$, and hyper-parameters. **Outputs:** A total of $T_{\mathrm{max}}$ jointly trained PGBNs with depths $T = 1$, $T = 2$, ..., and $T = T_{\mathrm{max}}$.

---

1: **for** $T = 1, 2, \ldots, T_{\mathrm{max}}$ **do** Jointly train all the $T$ layers of the network
2:      Set $K_{T-1}$, the inferred width of layer $T - 1$, as $K_{T\,\mathrm{max}}$, the upper bound of layer $T$'s width.
3:      **for** $iter = 1 : B_T + C_T$ **do** Upward-downward Gibbs sampling
4:          Sample $\{z_{ji}\}_{j,i}$ using collapsed inference; Calculate $\{x_{vjk}^{(1)}\}_{v,k,j}$; Sample $\{x_{vj}^{(2)}\}_{v,j}$ ;
5:          **for** $t = 2, 3, \ldots, T$ **do**
6:              Sample $\{x_{vjk}^{(t)}\}_{v,j,k}$ ; Sample $\{\phi_k^{(t)}\}_k$ ; Sample $\{x_{vj}^{(t+1)}\}_{v,j}$ ;
7:          **end for**
8:          Sample $p_j^{(2)}$ and Calculate $c_j^{(2)}$; Sample $\{c_j^{(t)}\}_{j,t}$ and Calculate $\{p_j^{(t)}\}_{j,t}$ for $t = 3, \ldots, T + 1$
9:          **for** $t = T, T - 1, \ldots, 2$ **do**
10:            Sample $\boldsymbol{r}$ if $t = T$; Sample $\{\boldsymbol{\theta}_j^{(t)}\}_j$ ;
11:          **end for**
12:          **if** $iter = B_T$ **then**
13:            Prune layer $T$'s inactive factors $\{\phi_k^{(T)}\}_{k:x_{\cdot\cdot k}^{(T)}=0}$, let $K_T = \sum_k \delta(x_{\cdot\cdot k}^{(T)} > 0)$, and update $\boldsymbol{r}$;
14:          **end if**
15:      **end for**
16:      Output the posterior means (according to the last MCMC sample) of all remaining factors $\{\phi_k^{(t)}\}_{k,t}$ as
       the inferred network of $T$ layers, and $\{r_k\}_{k=1}^{K_T}$ as the gamma shape parameters of layer $T$'s hidden units.
17: **end for**

---

$\mathrm{Gam}(\gamma_0/K_{T\,\mathrm{max}}, 1/c_0)$, and rely on that stochastic process's shrinkage mechanism to prune inactive factors (connection weight vectors) of layer $T$, and hence the inferred $K_T$ would be smaller than $K_{T\,\mathrm{max}}$ if $K_{T\,\mathrm{max}}$ is sufficiently large. The newly added layer and the layers below it would be jointly trained, but with the structure below the newly added layer kept unchanged. Note that when $T = 1$, the PGBN would infer the number of active factors if $K_{1\,\mathrm{max}}$ is set large enough, otherwise, it would still assign the factors with different weights $r_k$, but may not be able to prune any of them.

## 3 Experimental Results

We apply the PGBNs for topic modeling of text corpora, each document of which is represented as a term-frequency count vector. Note that the PGBN with a single hidden layer is identical to the (truncated) gamma-negative binomial process PFA of [13], which is a nonparametric Bayesian algorithm that performs similarly to the hierarchical Dirichlet process latent Dirichlet allocation [23] for text analysis, and is considered as a strong baseline that outperforms a large number of topic modeling algorithms. Thus we will focus on making comparison to the PGBN with a single layer, with its layer width set to be large to approximate the performance of the gamma-negative binomial process PFA. We evaluate the PGBNs' performance by examining both how well they unsupervisedly extract low-dimensional features for document classification, and how well they predict heldout word tokens. Matlab code will be available in http://mingyuanzhou.github.io/.

We use Algorithm 1 to learn, in a layer-wise manner, from the training data the weight matrices $\boldsymbol{\Phi}^{(1)}, \ldots, \boldsymbol{\Phi}^{(T_{\mathrm{max}})}$ and the top-layer hidden units' gamma shape parameters $\boldsymbol{r}$: to add layer $T$ to a previously trained network with $T - 1$ layers, we use $B_T$ iterations to jointly train $\boldsymbol{\Phi}^{(T)}$ and $\boldsymbol{r}$ together with $\{\boldsymbol{\Phi}^{(t)}\}_{1,T-1}$, prune the inactive factors of layer $T$, and continue the joint training with another $C_T$ iterations. We set the hyper-parameters as $a_0 = b_0 = 0.01$ and $e_0 = f_0 = 1$. Given the trained network, we apply the upward-downward Gibbs sampler to collect 500 MCMC samples after 500 burnins to estimate the posterior mean of the feature usage proportion vector $\boldsymbol{\theta}_j^{(1)}/\theta_{\cdot j}^{(1)}$ at the first hidden layer, for every document in both the training and testing sets.

**Feature learning for binary classification.** We consider the 20 newsgroups dataset (http://qwone.com/~jason/20Newsgroups/) that consists of 18,774 documents from 20 different news groups, with a vocabulary of size $K_0 = 61,188$. It is partitioned into a training set of 11,269 documents and a testing set of 7,505 ones. We first consider two binary classification tasks that distinguish between the $comp.sys.ibm.pc.hardware$ and $comp.sys.mac.hardware$, and between the $sci.electronics$ and $sci.med$ news groups. For each binary classification task, we remove a standard list of stop words and only consider the terms that appear at least five times, and report the classification accuracies based on 12 independent random trials. With the upper bound of the first layer's

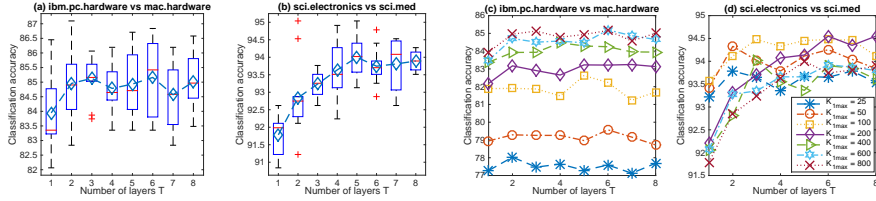

Figure 1: Classification accuracy (%) as a function of the network depth $T$ for two 20newsgroups binary classification tasks, with $\eta^{(t)} = 0.01$ for all layers. (a)-(b): the boxplots of the accuracies of 12 independent runs with $K_{1\max} = 800$. (c)-(d): the average accuracies of these 12 runs for various $K_{1\max}$ and $T$. Note that $K_{1\max} = 800$ is large enough to cover all active first-layer topics (inferred to be around 500 for both binary classification tasks), whereas all the first-layer topics would be used if $K_{1\max} = 25, 50, 100,$ or 200.

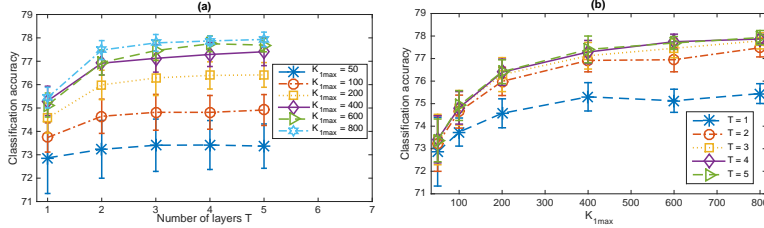

Figure 2: Classification accuracy (%) of the PGBNs for 20newsgroups multi-class classification (a) as a function of the depth $T$ with various $K_{1\max}$ and (b) as a function of $K_{1\max}$ with various depths, with $\eta^{(t)} = 0.05$ for all layers. The widths of hidden layers are automatically inferred, with $K_{1\max} = 50, 100, 200, 400, 600,$ or 800. Note that $K_{1\max} = 800$ is large enough to cover all active first-layer topics, whereas all the first-layer topics would be used if $K_{1\max} = 50, 100,$ or 200.

width set as $K_{1\max} \in \{25, 50, 100, 200, 400, 600, 800\}$, and $B_t = C_t = 1000$ and $\eta^{(t)} = 0.01$ for all $t$, we use Algorithm 1 to train a network with $T \in \{1, 2, \dots, 8\}$ layers. Denote $\bar{\boldsymbol{\theta}}_j$ as the estimated $K_1$ dimensional feature vector for document $j$, where $K_1 \leq K_{1\max}$ is the inferred number of active factors of the first layer that is bounded by the pre-specified truncation level $K_{1\max}$. We use the $L_2$ regularized logistic regression provided by the LIBLINEAR package [25] to train a linear classifier on $\bar{\boldsymbol{\theta}}_j$ in the training set and use it to classify $\bar{\boldsymbol{\theta}}_j$ in the test set, where the regularization parameter is five-folder cross-validated on the training set from $(2^{-10}, 2^{-9}, \dots, 2^{15})$.

As shown in Fig. 1, modifying the PGBN from a single-layer shallow network to a multi-layer deep one clearly improves the qualities of the unsupervisedly extracted feature vectors. In a random trial, with $K_{1\max} = 800$, we infer a network structure of $(K_1, \dots, K_8) = (512, 154, 75, 54, 47, 37, 34, 29)$ for the first binary classification task, and $(K_1, \dots, K_8) = (491, 143, 74, 49, 36, 32, 28, 26)$ for the second one. Figs. 1(c)-(d) also show that increasing the network depth in general improves the performance, but the first-layer width clearly plays an important role in controlling the ultimate network capacity. This insight is further illustrated below.

**Feature learning for multi-class classification.** We test the PGBNs for multi-class classification on 20newsgroups. After removing a standard list of stopwords and the terms that appear less than five times, we obtain a vocabulary with $K_0 = 33,420$. We set $C_t = 500$ and $\eta^{(t)} = 0.05$ for all $t$. If $K_{1\max} \leq 400$, we set $B_t = 1000$ for all $t$, otherwise we set $B_1 = 1000$ and $B_t = 500$ for $t \geq 2$. We use all 11,269 training documents to infer a set of networks with $T_{\max} \in \{1, \dots, 5\}$ and $K_{1\max} \in \{50, 100, 200, 400, 600, 800\}$, and mimic the same testing procedure used for binary classification to extract low-dimensional feature vectors, with which each testing document is classified to one of the 20 news groups using the $L_2$ regularized logistic regression. Fig. 2 shows a clear trend of improvement in classification accuracy by increasing the network depth with a limited first-layer width, or by increasing the upper bound of the width of the first layer with the depth fixed. For example, a single-layer PGBN with $K_{1\max} = 100$ could add one or more layers to slightly outperform a single-layer PGBN with $K_{1\max} = 200$, and a single-layer PGBN with $K_{1\max} = 200$ could add layers to clearly outperform a single-layer PGBN with $K_{1\max}$ as large as 800. We also note that each iteration of jointly training multiple layers costs moderately more than that of training a single layer, e.g., with $K_{1\max} = 400$, a training iteration on a single core of an Intel Xeon 2.7 GHz CPU on average takes about 5.6, 6.7, 7.1 seconds for the PGBN with 1, 3, and 5 layers, respectively.

Examining the inferred network structure also reveals interesting details. For example, in a random trial with Algorithm 1, the inferred network widths $(K_1, \dots, K_5)$ are

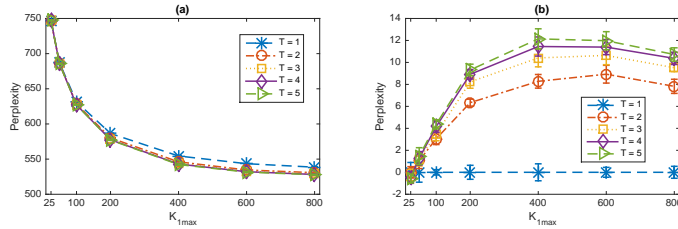

Figure 3: (a) per-heldout-word perplexity (the lower the better) for the NIPS12 corpus (using the 2000 most frequent terms) as a function of the upper bound of the first layer width $K_{1\max}$ and network depth $T$, with 30% of the word tokens in each document used for training and $\eta^{(t)} = 0.05$ for all $t$. (b) for visualization, each curve in (a) is reproduced by subtracting its values from the average perplexity of the single-layer network.

$(50, 50, 50, 50, 50)$, $(200, 161, 130, 94, 63)$, $(528, 129, 109, 98, 91)$, and $(608, 100, 99, 96, 89)$, for $K_{1\max} = 50, 200, 600,$ and $800$, respectively. This indicates that for a network with an insufficient budget on its first-layer width, as the network depth increases, its inferred layer widths decay more slowly than a network with a sufficient or surplus budget on its first-layer width; and a network with a surplus budget on its first-layer width may only need relatively small widths for its higher hidden layers. In the Appendix, we provide comparisons of accuracies between the PGBN and other related algorithms, including these of [9] and [26], on similar multi-class document classification tasks.

**Perplexities for holdout words.** In addition to examining the performance of the PGBN for unsupervised feature learning, we also consider a more direct approach that we randomly choose 30% of the word tokens in each document as training, and use the remaining ones to calculate per-heldout-word perplexity. We consider the NIPS12 (http://www.cs.nyu.edu/~roweis/data.html) corpus, limiting the vocabulary to the 2000 most frequent terms. We set $\eta^{(t)} = 0.05$ and $C_t = 500$ for all $t$, set $B_1 = 1000$ and $B_t = 500$ for $t \geq 2$, and consider five random trials. Among the $B_t + C_t$ Gibbs sampling iterations used to train layer $t$, we collect one sample per five iterations during the last $500$ iterations, for each of which we draw the topics $\{\phi_k^{(1)}\}_k$ and topics weights $\theta_j^{(1)}$, to compute the per-heldout-word perplexity using Equation (34) of [13]. As shown in Fig. 3, we observe a clear trend of improvement by increasing both $K_{1\max}$ and $T$.

**Qualitative analysis and document simulation.** In addition to these quantitative experiments, we have also examined the topics learned at each layer. We use $\left( \prod_{\ell=1}^{t-1} \Phi^{(\ell)} \right) \phi_k^{(t)}$ to project topic $k$ of layer $t$ as a $V$-dimensional word probability vector. Generally speaking, the topics at lower layers are more specific, whereas those at higher layers are more general. E.g., examining the results used to produce Fig. 3, with $K_{1\max} = 200$ and $T = 5$, the PGBN infers a network with $(K_1, \ldots, K_5) = (200, 164, 106, 60, 42)$. The ranks (by popularity) and top five words of three example topics for layer $T = 5$ are "6 network units input learning training," "15 data model learning set image," and "34 network learning model input neural;" while these of five example topics of layer $T = 1$ are "19 likelihood em mixture parameters data," "37 bayesian posterior prior log evidence," "62 variables belief networks conditional inference," "126 boltzmann binary machine energy hinton," and "127 speech speaker acoustic vowel phonetic." We have also tried drawing $\theta^{(T)} \sim \text{Gam}\left(r, 1/c_j^{(T+1)}\right)$ and downward passing it through the $T$-layer network to generate synthetic documents, which are found to be quite interpretable and reflect various general aspects of the corpus used to train the network. We provide in the Appendix a number of synthetic documents generated from a PGBN trained on the 20newsgroups corpus, whose inferred structure is $(K_1, \ldots, K_5) = (608, 100, 99, 96, 89)$.

# 4 Conclusions

The Poisson gamma belief network is proposed to extract a multilayer deep representation for high-dimensional count vectors, with an efficient upward-downward Gibbs sampler to jointly train all its layers and a layer-wise training strategy to automatically infer the network structure. Example results clearly demonstrate the advantages of deep topic models. For big data problems, in practice one may rarely has a sufficient budget to allow the first-layer width to grow without bound, thus it is natural to consider a belief network that can use a deep representation to not only enhance its representation power, but also better allocate its computational resource. Our algorithm achieves a good compromise between the widths of hidden layers and the depth of the network.

**Acknowledgements.** M. Zhou thanks TACC for computational support. B. Chen thanks the support of the Thousand Young Talent Program of China, NSC-China (61372132), and NCET-13-0945.

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
