[Supplementary Material]

# Appendix for The Poisson Gamma Belief Network

## A    Comparisons of classification accuracies

For comparison, we consider the same $L_2$ regularized logistic regression multi-class classifier, trained either on the raw word counts or normalized term-frequencies of the 20newsgroups training documents using five-folder cross-validation. As summarized in Tab. 1, when using the raw term-frequency word counts as covariates, the same classifier achieves 69.8% (68.2%) accuracy on the 20newsgroups test documents if using the top 2000 terms that exclude (include) a standard list of stopwords, achieves 75.8% if using all the $61,188$ terms in the vocabulary, and achieves 78.0% if using the $33,420$ terms remained after removing a standard list of stopwords and the terms that appear less than five times; and when using the normalized term-frequencies as covariates, the corresponding accuracies are 70.8% (67.9%) if using the top 2000 terms excluding (including) stopwords, 77.6% with all the $61,188$ terms, and 79.4% with the $33,420$ selected terms.

Table 1: Multi-class classification accuracy of $L_2$ regularized logistic regression.

| $V = 61,188$ with stopwords with rare words raw word counts | $V = 61,188$ with stopwords with rare words term frequencies | $V = 33,420$ remove stopwords remove rare words raw word counts | $V = 33,420$ remove stopwords remove rare words term frequencies |
|---|---|---|---|
| 75.8% | 77.6% | 78.0% | 79.4% |

| $V = 2000$ with stopwords raw counts | $V = 2000$ with stopwords term frequencies | $V = 2000$ remove stopwords raw counts | $V = 2000$ remove stopwords term frequencies |
|---|---|---|---|
| 68.2% | 67.9% | 69.8% | 70.8% |

As summarized in Tab. 2, for multi-class classification on the same dataset, with a vocabulary size of 2000 that consisits of the 2000 most frequent terms after removing stopwords and stemming, the DocNADE [9] and the over-replicated softmax [26] provide the accuracies of 67.0% and 66.8%, respectively, for a feature dimension of $K = 128$, and provide the accuracies of 68.4% and 69.1%, respectively, for a feature dimension of $K = 512$.

Table 2: Multi-class classification accuracy of the DocNADE [9] and over-replicated softmax [26].

|  | $V = 2000, K = 128$ remove stopwords, stemming | $V = 2000, K = 512$ remove stopwords, stemming |
|---|---|---|
| DocNADE | 67.0% | 68.4% |
| Over-replicated softmax | 66.8% | 69.1% |

As summarized in Tab. 3, with the same vocabulary size of 2000 (but different terms due to different preprocessing), the proposed PGBN provides 65.9% (67.5%) with $T = 1$ ($T = 5$) for $K_{1\max} = 128$, and 65.9% (69.2%) with $T = 1$ ($T = 5$) for $K_{1\max} = 512$, which may be further improved if we also consider the stemming step, as done in the these two algorithms, for word preprocessing, or if we set the values of $\eta^{(t)}$ to be smaller than 0.05. We also summarize in Tab. 3 the classification accuracies of the PGBNs learned with $V = 33,420$, as shown in Fig. 2.

## B    Generating synthetic documents

Below we provide several synthetic documents generated from the PGBN with $(K_1, \ldots, K_5) = (608, 100, 99, 96, 89)$, which is trained on the training set of the 20newsgroups corpus with $K_{1\max} = 800$ and $\eta^{(t)} = 0.05$ for all $t$. We set $c_{j'}^{(t)}$ as the median of the inferred $\{c_j^t\}_j$ of the training documents for all $t$. Given $\{\mathbf{\Phi}^{(t)}\}_{1,T}$ and $\boldsymbol{r}$, We first generate $\boldsymbol{\theta}_{j'}^{(T)} \sim \text{Gam}\left(\boldsymbol{r}, 1/c_{j'}^{(T+1)}\right)$

Table 3: Classification accuracy of the PGBN trained with $\eta^t = 0.05$ for all $t$.

|  | $V = 2000, K_{1\max} = 128$ remove stopwords | $V = 2000, K_{1\max} = 256$ remove stopwords | $V = 2000, K_{1\max} = 512$ remove stopwords |
|---|---|---|---|
| PGBN ($T=1$) | $65.9\% \pm 0.4\%$ | $66.3\% \pm 0.4\%$ | $65.9\% \pm 0.4\%$ |
| PGBN ($T=5$) | $67.5\% \pm 0.4\%$ | $68.8\% \pm 0.3\%$ | $69.2\% \pm 0.4\%$ |

|  | $V = 33,420, K_{1\max} = 200$ remove stopwords remove rare words | $V = 33,420, K_{1\max} = 400$ remove stopwords remove rare words | $V = 33,420, K_{1\max} = 800$ remove stopwords remove rare words |
|---|---|---|---|
| PGBN ($T=1$) | $74.6\% \pm 0.6\%$ | $75.3\% \pm 0.6\%$ | $75.4\% \pm 0.4\%$ |
| PGBN ($T=5$) | $76.4\% \pm 0.5\%$ | $77.4\% \pm 0.6\%$ | $77.9\% \pm 0.3\%$ |

and then downward pass it through the network by repeatedly drawing nonnegative real random variables from the gamma distribution as in (1). With the simulated $\boldsymbol{\theta}_{j'}^{(1)}$, we calculate the Poisson rates for all the $V$ words using $\boldsymbol{\Phi}^{(1)}\boldsymbol{\theta}_{j'}^{(1)}$ and display the top 100 words ranked according to $\boldsymbol{\Phi}^{(1)}\boldsymbol{\theta}_{j'}^{(1)}$. Below are some example synthetic documents generated in this manner, which are all easy to interpret and reflect various aspects of the 20newsgroups corpus used to train the PGBN.

- team game games hockey year cup season playoffs edu win pittsburgh nhl toronto detroit stanley teams montreal play jets pens espn division chicago new penguins pick league players devils rangers wings boston islanders playoff ca series winnipeg gm abc tv playing quebec april time round st vancouver fans best gld bruins coach winner calgary leafs player great watch night patrick vs finals conference final just baseball coverage murray minnesota don won gary points mike like ice kings regular mario played louis caps contact washington selanne norris buffalo columbia keenan star people fan th think canadiens said canada canucks york gerald

- hall smith players fame career ozzie winfield nolan guys ryan dave baseball eddie murray numbers steve kingman robinson yount morris roger years bsu puckett long joe jackson hung brett garvey deserve robin evans princeton yeah frank ruth kirby rickey pitcher peak yogi hof great sick lee ha aaron johnny darrell santo time greatest stats seasons ron george reardon shortstops henderson hank mays jack liability marginal rogers average compare belong schmidt gibson willie leo ucs sgi bsuvc comment fans honestly deserves cal bell candidates wagner fielding walks ve likely history gee heck consideration mike player bonds lock rating sandberg standards apparent

- fbi koresh batf gas compound waco government atf people children tear cult davidians did bd branch agents happened assault warrant david reno tanks killed weapons clinton point country search building federal raid press started reported death proper needed illegal better house protect burned janet outside burn days media stand job arms inside right come cwru equipment followers investigation oldham believe non power kids burning fires women suicide law order cs sick blame initial alive feds agent tank religious automatic davidian deaths knock good hit said military possible died away light fault child witnesses pay instead folks daniel bureau armored going

- people government law state israel rights israeli jews right public states war fact political country arab laws article case court human federal american united support society policy civil freedom members national jewish evidence person majority force power legal citizens action crime world act countries issue arabs group police justice non control palestinian live land peace true anti center writes gaza population research constitution death edu org allowed party protection consider actions number adam apc general subject based murder igc considered life military self parties lives personal nation order cpr social question individual religious today situation free responsibility governments palestine innocent

- medical health disease doctor pain patients treatment medicine cancer edu hiv blood use years patient writes cause skin don like just aids symptoms number article help diseases drug com effects information doctors infection physician normal chronic think taking care volume condition drugs page says cure people tobacco hicnet know newsletter effective therapy problem common time women prevent surgery children center immune research

called april control effect weeks low syndrome hospital physicians states clinical diagnosed day med age good make caused severe reported public safety child said cdc usually diet national studies tissue months way cases causing migraine smokeless infections does

- card video drivers cards driver vga mode ati graphics windows diamond vesa bus svga support gateway dx pc modes color isa board version local bit memory vlb ultra pro eisa monitor new does mb stealth hz using based speedstar orchid colors available latest ram know work chip performance resolution fast screen speed tech million trident winbench dcoleman set problems yes et ftp results winmarks plus edu bbs zeos utexas vram bios robert win higher magazine utxvms able high interlaced viper com boards site weitek tseng chipset modem turbo software non resolutions far faster accelerated supports price meg ega mhz true

- card windows video drivers monitor com modem vga cards driver port pc mode screen ati serial graphics dos bus board irq support svga diamond vesa using memory problem dx color gateway file version ports local modes pro bit does isa colors mb know vlb mouse ultra win ram new monitors hz work eisa nec problems chip files stealth use set program speedstar orchid plus high based resolution fast software cable hardware display latest used performance ms like baud bbs tech connector run thanks speed just yes million trident winbench dcoleman available pin ibm uart connect sony window switch et disk

- nissan electronics wagon altima delcoelect kocrsv station gm subaru sumax delco spiros hughes wax pathfinder legacy kokomo wagons smorris scott toyota seattleu don just like strong silver software luxury derek proof stanza seattle cisco morris cymbal triantafyl-lopoulos sportscar think people know near fool ugly proud claims flat statistics lincoln sedans bullet karl lee perth puzzled miata sentra maxima acura infiniti corolla mgb untruth verbatim good time consider way based make stand guys writes noticed want ve heavy suggestion eat steven horrible uunet studies armor fisher lust designs study definately lexus remove conversion embodied aesthetic elvis attached honey stole designing wd

- mac apple bit mhz ram simms mb like memory just don cpu people chip chips think color board ibm speed does know se video time machines motherboard hardware lc cache meg ns simm need upgrade built vram good quadra want centris price dx run way processor card clock slots make fpu internal did macs cards ve pin power really machine say faster said software intel macintosh right week writes slot going sx performance things edu years nubus possible thing monitor work point expansion rom iisi ll add dram better little slow let sure pc ii didn ethernet lciii case kind

- image jpeg gif file color files images format bit display convert quality formats colors pro-grams program tiff picture viewer graphics bmp bits xv screen pixel read compression conversion zip shareware scale view jpg original save quicktime jfif free version best pcx viewing bitmap gifs simtel viewers don mac usenet resolution animation menu scanner pix-els sites gray quantization displays better try msdos tga want current black faq converting white setting mirror xloadimage section ppm fractal amiga write algorithm mpeg pict targa arithmetic export scodal archive converted grasp lossless let space human grey directory pictures rgb demo scanned old choice grayscale compress

- gun guns edu writes bike com article weapons dod control crime weapon apr used carry criminals police ride nra bikes self firearms use buy firearm laws concealed bmw defense home handgun criminal motorcycle anti problem car people owners ban rider riding shot just armed new don like crimes assault kill violent protect uio handguns ifi evil ama citizens state org know illegal politics texas thomas thomasp cb talk legal shooting pro road carrying abiding think att honda cs stolen defend good purchase ll law individual hp cc permit rifle issue government states parsli property ve killing federal does motorcycles time

- gun guns weapons people control government law crime state rights police laws weapon self criminals carry states public nra used defense firearms anti federal right criminal legal firearm citizens country home political case concealed handgun court fact crimes issue protect armed politics kill ban problem buy national individual support shot society violent use civil war property talk owners assault illegal handguns ifi uio united defend action allowed freedom article american amendment person member power force thomasp car human evidence threat thomas murder shooting majority killed carrying members citizen killing pro abiding group act evil texas america justice permit stolen said