[Reviews · NeurIPS 2015]

Submitted by Assigned_Reviewer_1

The authors propose a multilayer representation of count vectors using a hierarchical model. The model has hidden layers consisting of gamma variables drawn from distributions with factorized shape parameters.

The proposed Gibbs sampling based inference is capable of learning the widths of the hidden layers, with the first-layer width acting as limit. Using this, the experimental results show that a deeper network has better classification accuracy and perplexity than a single-layer model (which is equivalent to the model of [12]). The superior performance is attributed to capturing correlations between hidden units and modeling overdispersion.

The model and algorithm are clearly presented.

The paper draws out the advantage of the multilayer over the single layer model -- equivalent to a nonparametric extension to the PFA presented in Zhou et al. (2015) -- and the experimental results are convincing.

While the model is novel -- using nonnegative hidden units instead of binary, and automatically learning the widths -- the advantage over the leading competing method, that of the over-replicated softmax [21] -- a two layer DBM model -- is not addressed in any depth. The authors note that

the classification accuracy is worse than the over-replicated softmax [21] when the first-layer budget is 512. This is noted to possibly due to word preprocessing.

Further, on pg 8., the authors draw observations about layer width decay rates, but these come from inference on a single data set. To what extent is this a function of the data vs. the budget of the first-layer? The claim on pg.2 that you reveal this relationship surely needs analysis on more than 1 dataset.

Summary: A novel multilayer model of count vectors with nonnegative hidden units and Gibbs sampling-based inference is proposed. The merits over a (possibly wider) single-layer model has been clearly presented, and the experimental results are convincing. However, the paper does not demonstrate that the model is the state-of-the-art in a domain (e.g., topic modeling); neither does it demonstrate merits over the leading competing method -- the overreplicated softmax model of Srivastava et al. (2013), a deep Bolztman machine with binary hidden units.

Further, the authors draw conclusions from results on a single data set.

Submitted by Assigned_Reviewer_2

Acknowledge the author's rebuttal.

I overall maintain my general sentiment about the paper and look forward to more discussion of this work's relationship to traditional neural / deep networks.

---

In this paper the authors present a deep belief network in which the intermediate hidden layers are represented by nonnegative weights.

They apply this model to text documents as a "deep belief topic model" (my own phrasing).

They derive an inference / update step, in which dirichlet vectors are propagated up the network and gamma weights are propagated back down it, and perform an empirical evaluation of the model.

Overall this paper was written fairly clearly.

I thought the paper was mostly fine, but I believe that the empirical analysis could be improved (more on that shortly).

I also would be shocked if nobody has tried to implement a neural network with weights outside the range [0, 1] before; it would be good to see some background material discussed even if it wasn't used to motivate this model.

I also had a few other minor comments:

- It's not clear whether a fixed budget is appropriate for fixing the size of the various layers (and, in the absence of more detail about the method, I am skeptical).

It still seems that these layers could easily be too large or too small despite the fixed budget approach.

I suspect that some held-out subset of data could be used to decide whether to grow or shrink the model.

- It's not clear whether adding any layers beyond the second layer help.

The performance in the figures indicates that they're marginally better, but it's difficult to be confident given how much variance there is in the plots (with the possible exception of Figure 3).

Further, the topic descriptions from sample words (line 417) aren't substantially different; it's very possible that the authors' observation that the topics are more specific at the top layer are simply because that layer is largest.

- The authors describe a process to jointly train all layers in contrast to training in a greedy fashion (line 264).

In this respect it resembles the way a traditional neural network (from 20 years ago) was fit, i.e., forward/back propagation, except that it is with samples.

- It seems that this paper could introduce the idea of the Gamma-Poisson belief network without needing to use an application which uses the CRT (i.e., the application itself is complicated, and it's not clear that this paper necessitates a complicated application to evaluate a neural network architecture.

Regarding the experimental validation, I believe it could be improved.

First, it would be useful to see more comparisons with baselines.

For the classification task, for example, it would be good to see an SVM or ridge regression using word counts as covariates, which I have seen outcompete topic models for classification. (N.B. I don't think the model needs to beat these baselines).

It would be good to see some comparison with e.g. a typical neural network, in which weights are in [0,1], perhaps with the bottom layer adapted to suit the Poisson/Multinomial observations.

Particularly, claims like that on 145 ("..clearly shows that the gamma distributed nonnegative hidden units could carry richer information than the binary hidden units and model more complex nonlinear functions") could an should be backed up with a comparison with a model using binary hidden units.

It's not clear that a binary network couldn't express similar nonlinear functions with the right architecture.

Is it necessary to remove stopwords?

Topic models like LDA can handle them fine if you're okay with a stopword topic (and it's not clear why that would hurt an evaluation).

Nit picks:

- The authors discuss classification in the experiment section (line 349) before the actual task is introduced in 353, which was a bit confusing.

- Line 56: "budge" -> "budget"
Summary: This paper offers an interesting variant of a typical deep network.

The idea is interesting, and the presentation is mostly clear, but the experimental validation could be improved.

Submitted by Assigned_Reviewer_3

Summary: A poisson gamma hierarchy is presented for modeling counts data.

The primary contribution lies in utilizing a hierarchy of gamma distributions and using recently proposed clever augmentation tricks to develop a simple, tractable Gibbs sampler in spite of non conjugacy. Coupled with a Poisson likelihood, the authors demonstrate the utility of the hierarchy for modeling counts data.

The paper is technically sound. The problem of extracting unsupervised multilayered representations of data is interesting, and the developed model provides an interesting alternative for counts data.

Detailed comments:

1) While it is true that RBMs have traditionally used binary hidden units, recent work [1] has found that using rectified linear nonlinearities leads to better representations. The non linearities induced by the proposed gamma units appear to subsume the linear regime of rectified linear nonlinearity (with the linear regime being recovered in expectation, when the expected rate parameter is 1). This is cool and it would be nice to discuss this connection more explicitly in the text.

2) Scalability appears to be a big concern for the inference procedure. It isn't completely clear, but it appears that a 1000 Gibbs sampling iterations are run after adding each new layer, sweeping over all the variables in the network. This would be difficult to scale to deeper architectures. How long does it currently take to train the 5 layer networks employed for multi-class classification? A discussion of these computational issues would be useful.

3) The external classification comparisons, to DocNADE and over replicated softmax are sloppy. The classification numbers are not really comparable, considering the competing algorithms are trained on distinct vocabularies. The paragraph on qualitative analysis is not useful at all to the reader and should be reworked. The authors claim that the discovered topics specialize as one traverses the hierarchy downwards, but present very little evidence in support of this claim. It would be interesting for the reader to explore the per layer discovered topics more closely and they could be made available in a supplement. The authors also claim, without supporting

evidence, that they can generate interpretable synthetic documents from the trained network. It would be interesting to see these synthetic documents generated by the network. Again something that could be easily provided in an appendix.

[1] Nair, Vinod, and Geoffrey E. Hinton. "Rectified linear units improve restricted boltzmann machines." Proceedings of the 27th International Conference on Machine Learning (ICML-10). 2010.
Summary: Overall, this is an interesting paper. I would have scored the paper higher, if not for sloppy experimental evaluations.

Author Feedback
Author rebuttal: We will add more qualitative analysis given the reviewers' clear interest. We will share the code in Github.

Below are some additional NIPS12 topics. The topics of the bottom (top) layer are very specific (general). E.g., Topic 41 of the bottom layer is about music & protein!

For the bottom layer:
12: speech recognition word system hmm
20: regression functions basis rbf local
39: risk market stock financial return
41: music structure protein amino concert

For the top layer:
6: network training neural speech learning
14: data set function learning algorithm
23: network neural learning input function

Below are the top 80 words of a synthetic document generated by the PGBN trained on 20newsgroups:

game play period team goal power flyers pittsburgh puck pp toronto season chicago calgary pts kings st detroit buffalo win montreal second islanders scoring points goals new shots boston year games devils cup winnipeg rangers penalty blues scorer vancouver captain shot louis pens flames traded quebec hartford played lemieux got division jersey mike pick penguins leafs jets series philadelphia lindros winner penalties washington net ny angeles soderstrom nd recchi th playoffs los rd beat stanley record vs goalie line sabres

Clearly, it is about Ice Hockey. E.g., Calgary Flames and Philadelphia Flyers are NHL teams; Lemieux, Lindros and Soderstrom are NHL stars; and power play, puck, goalie are Ice Hockey terms.

The accuracies in Lines 404-408 were provided to show that our models and the DocNADE [9] and overreplicated softmax [21] were in the same ballpark. They were not intended to provide a head-to-head comparison, which was impossible for us to do since the code, preprocessed data and error bars were not available for [21]. Nevertheless, since [21] did both stopwords removal and stemming, whereas we only removed stopwords, we might achieve better results given the same data preprocessing.

To Reviewer 1: We disagree with your three criticisms. First, the proposed PGBN clearly outperforms the gamma-negative binomial process topic model, a state-of-the-art nonparametric Bayesian algorithm. Please see our response above to your second criticism. Third, we drew observations on layer width decay rates on not only NIPS12 but also 20newsgroups (Lines 416 and 390), and also made related comments in Figs. 1-2's Captions.

To Reviewer 2: For previous work on non-binary units, we will make connections to the rectified linear units of Vinod & Hinton, 2010, and add comments on "Ranganath, Tang, Charlin & Blei, deep exponential families, AISTATS 2015."

Given more data, our nonparametric Bayesian algorithm could automatically adjust its layer widths to avoid under fitting.

In general, the gain of adding a layer gradually diminishes as the network depth increases, and the gains are often clearer for larger and more complex data. We observe similar general vs specific distinctions when the inferred layer widths are similar. Please see our response in the beginning for example topics of the bottom and top layers.

For unsupervised learning, existing deep models might fail without greedy layer-wise training, except for the deep Boltzmann machine with approximate inference. Our deep Bayesian model can jointly train all its layers in an unsupervised manner, without using mean-field approximation. The CRT is introduced to facilitate inference.

Linear SVM using the raw word counts as covariates achieves 69.8% (68.1%) accuracy on 20newsgroups if we use the top 2000 terms with (without) stopwords removal, and 75.8% if we use all the terms. We will include these results as baselines.

The overreplicated softmax [21] seems to match your suggested model. As a head-to-head comparison with [21] was impossible (please see our response in the beginning), we will focus on presenting the gamma units as a new choice, and tone down our argument that they could carry more information than binary ones.

Removing stopwords often helps improve topic interpretability. For feature extraction for classification, our additional experiments do not show that it is necessary to remove stopwords. Thank you for "nit picks"!

To Reviewer 3: Thank you for suggesting the interesting connections between our gamma units and the rectified linear units of Vinod & Hinton, 2010. We will cite and make connections to that paper.

Please see our previous response on comparisons and qualitative analysis.

To Reviewers 3-4: We will discuss more about computation. A nice property: jointly training a five-layer model costs moderately more than training a single-layer one. E.g., for 20newsgroups with K_1max=128, a training iteration on a 3.4GHz CPU takes about 1.32, 1.85 and 2.08 seconds for a PGBN model with 1, 3 and 5 layers, respectively.

To Reviewer 5: We believe a new deep network structure distinct from existing ones would be of interest to both the deep learning and Bayesian machine learning communities.